# Assessing Enacted Sexual Stigma toward Gay and Bisexual Men in the Military: The Enacted Sexual Stigma Experiences Scale in Military Service

**DOI:** 10.3390/ijerph20021260

**Published:** 2023-01-10

**Authors:** Chung-Ying Lin, Yu-Ping Chang, Wen-Jiun Chou, Cheng-Fang Yen

**Affiliations:** 1Institute of Allied Health Sciences, College of Medicine, National Cheng Kung University, Tainan 70101, Taiwan; 2Biostatistics Consulting Center, National Cheng Kung University Hospital, College of Medicine, National Cheng Kung University, Tainan 70101, Taiwan; 3Department of Public Health, College of Medicine, National Cheng Kung University, Tainan 70101, Taiwan; 4Department of Occupational Therapy, College of Medicine, National Cheng Kung University, Tainan 70101, Taiwan; 5School of Nursing, The State University of New York, University at Buffalo, Buffalo, NY 14260, USA; 6Department of Child and Adolescent Psychiatry, Chang Gung Memorial Hospital, Kaohsiung Medical Center, Kaohsiung 83301, Taiwan; 7School of Medicine, Chang Gung University, Taoyuan 33302, Taiwan; 8Department of Psychiatry, School of Medicine, College of Medicine, Kaohsiung Medical University, Kaohsiung 80708, Taiwan; 9Department of Psychiatry, Kaohsiung Medical University Hospital, Kaohsiung 80756, Taiwan; 10College of Professional Studies, National Pingtung University of Science and Technology, Pingtung 91201, Taiwan

**Keywords:** bisexual, gay, instrument, military, psychological well-being, stigma

## Abstract

Gay and bisexual military servicemembers experience disproportionately high rates of victimization due to enacted sexual stigma (ESS). This study formulated a new scale, called the Enacted Sexual Stigma Experiences Scale in Military Service (ESSESiMS) for gay and bisexual servicemembers, and examined its psychometric propensities. The five-item ESSESiMS was first developed based on the results of focus group interviews with 12 participants. A total of 399 gay and bisexual men who have experience of the military service participated in the study. Exploratory factor analysis (EFA) was implemented to determine the factor structure of the ESSESiMS; the internal consistency and concurrent validity of the ESSESiMS was also examined. The EFA results indicate that the ESSESiMS should have a single-item structure. The ESSESiMS exhibited acceptable internal consistency and concurrent validity. Incidents of ESS in the ESSESiMS were significantly associated with perceived sexual stigma outside the military service and with higher levels of depression, loneliness, and suicidal ideation. The results of our study supported the psychometric properties of the newly developed ESSESiMS for assessing the experiences of ESS among gay and bisexual servicemembers in Taiwan. Experiences of ESS toward gay and bisexual servicemembers were common; ESS was significantly associated with adverse mental health outcomes.

## 1. Introduction

Enacted sexual stigma (ESS) is a type of sexual stigma characterized by overt discriminatory behaviors against lesbian, gay, and bisexual (LGB) individuals [1]. LGB military servicemembers experience disproportionately high rates of victimization due to ESS [2,3]. For example, a study on 71,570 servicemembers in the active-duty military of the United States (U.S.) in 2000 revealed that 37% had witnessed or experienced harassment and violence based on perceptions or suspicions that the victim was gay; physical assault was reported by 5.3% of the respondents [4]. Another U.S. study on 208 LGB servicemembers in 2010 reported that 29% of the participants have been teased or mocked and 7% have received threats or injuries because of their sexual orientation [5]. In addition, a U.S. study on 253 LGB servicemembers in 2013 also reported that 83% of lesbian and bisexual women and 74% of gay and bisexual men experienced sexual harassment or assault in the past 12 months [3]. The results of previous studies have supported that ESS is a very common issue in the military service that needs attention.

According to minority stress theory [6], ESS can compromise the mental and physical health of LGB servicemembers [7,8]. Studies have indicated that LGB servicemembers and veterans have a higher risk of suicide [9,10], substance use [9,11,12,13], post-traumatic stress disorder [12,14], depression [12,14,15], anxiety [12], multiple somatic symptoms [12], sexually transmitted infections [9], and insomnia [12] compared with their heterosexual counterparts. However, few studies have examined the experiences of ESS among LGB servicemembers [7], especially in countries outside North America and Europe. Research has found that sexual orientation-based harassment is significantly associated with decreased social cohesion among U.S. military veterans [16]; decreased social support can further increase the risk of suicidal ideation [17]. Sexual assault during military service is significantly associated with the risk of post-traumatic stress disorder (PTSD) in lesbian and bisexual female veterans [18]. Concealment of sexual orientation is a common method for LGB servicemembers to avoid ESS in the military service; however, anxiety around the concealment of one’s sexual orientation while in the service was related to current depression and PTSD symptoms [14]. Further study on the association between ESS and mental health problems is needed to assist LGB servicemembers and prevent sexual orientation-based harassment [2].

Incidents of ESS in the military service not only harm the health of LGB servicemembers but also demoralize other LGB servicemembers who are not directly victimized [2]. However, as with other types of sexual stigma, ESS is underreported by those experiencing discrimination, bystanders, and military supervisors [3]. The use of a self-reported survey instrument can not only record incidents of ESS but also help foster a military culture that accepts and integrates LGB servicemembers [7]. Estrada et al. developed an 8-item sexual orientation harassment scale (SOHS) to assess LGB servicemembers’ experiences of harassment, including offensive speech, offensive/hostile gestures, threats or intimidation, graffiti, vandalism, physical assault, limited career opportunities, and unwarranted discipline or punishment [19]. Although the SOHS serves as an instrument for assessing LGB servicemembers’ experiences of harassment, a new scale is required for the assessments of ESS among LGB servicemembers in Asia for the following reasons. First, several Asian countries have a two-tiered military structure: one for volunteer soldiers, and the other for conscripts. For example, in Taiwan, men who are 18 years or older must complete at least 4 months of compulsory military service and adult men can also voluntarily enlist in the military. Therefore, the item “limited career opportunities” on the SOHS does not apply to gay and bisexual male servicemembers in compulsory military service. Second, the ESS directed toward LGB individuals may differ in presentation due to sociocultural differences among societies [20]. For example, graffiti, one item on the SOHS, is rare in Taiwanese military camps. Third, studies have demonstrated that sexual stigma from the public, family members, and peers is significantly associated with adverse mental health outcomes in gay and bisexual men (GBM) [21,22]. Whether experiences of ESS as measured by a standardized scale are related to adverse mental health outcomes in GBM in the military requires further study.

The Enacted Sexual Stigma Experiences Scale in Military Service (ESSESiMS) is a newly formulated scale assessing the experience of ESS among gay and bisexual servicemembers in Taiwan. This present study aimed to examine the psychometrics of the ESSESiMS. In addition to examining the factor structure of the ESSESiMS, this study also examined the concurrent validity of the ESSESiMS by testing its correlations with perceived sexual stigma outside military services, depression, loneliness, and suicidal ideation in gay and bisexual servicemembers. According to ecological systems theory [23], the army is an ecological system interacting with other systems. Therefore, it is hypothesized that the ESS measured by the ESSESiMS is significantly associated with the perceived sexual stigma outside military services. Moreover, according to minority stress theory [6], ESS constitutes a psychological stressor for GBM and negatively affects their mental health. Therefore, it is hypothesized that the ESS measured by the ESSESiMS is significantly associated with depression, loneliness, and suicidality.

## 2. Materials and Methods

### 2.1. Participants and Procedure

This present study took a two-stage approach to developing the ESSESiMS and examining its psychometric characteristics. Firstly, the investigators collected data on incidents of ESS in the military service from 12 GBM in 2 focus group interviews to help develop the ESSESiMS in January and February 2021. This study recruited participants by posting advertisements on the home pages of three online LGB communities. GBM with at least 4 months of military service experience in Taiwan were eligible to participate in the focus group interviews. The investigators determined the discussion topics and led the group discussion on incidents of ESS that the participants had experienced or witnessed during military service. Two researchers reviewed the transcript and coded the data for ESS in the military service. The principal investigator reviewed the coding results and integrated them into the following five items of the ESSESiMS: verbal harassment or social exclusion, physical violence (e.g., pushing, kicking, slamming, or other physical attack), unfair assignment of work, sexual harassment or abuse, and being rejected from receiving supervisors’ help. Each item was answered with “yes” or “no.”

Secondly, this study recruited participants to validate the ESSESiMS through advertisements on social media (Facebook, LINE, and Twitter) and on a popular forum in Taiwan, called the Bulletin Board System, from 1 August 2021 to 1 May 2022. This study included Taiwanese men who (1) identified as gay or bisexual, (2) were ≥20 years of age, and (3) were currently completing or had previously completed military service in Taiwan. This study excluded individuals with any conditions that prevented them from completing the questionnaire, such as those with cognitive impairment or dysfunction due to major physical or psychiatric disorders. Individuals who were interested in this study could contact research assistants by telephone. Research assistants tentatively evaluated the eligibility of potential participants, explained the study aims and procedures, and scheduled a time for eligible participants to complete the study questionnaires. A total of 401 potential participants were invited to participate in this study. The research assistants further determined whether they had conditions that might compromise their ability to complete the questionnaire in interview rooms of a university-affiliated hospital. Two individuals were excluded due to low mentality or alcohol intoxication. A total of 399 eligible GBM granted informed consent and individually completed questionnaires. This study was approved by the institutional review board of a university-affiliated hospital (KMUHIRB-F(I)-20210003).

### 2.2. Measures

#### 2.2.1. Demographic Characteristics

This present study asked the participants to report their age, educational level, sexual orientation, and duration of military service and whether they identified as being transgender (Table 1). There were no significant differences in age, sexual orientation, education level, and duration in the military service between the participants of the first and second stages. No participant in the first stage had a transgender identity.

#### 2.2.2. Sexual Stigma Outside Military Services

This study implemented the homosexuality-related stigma scale (HRSS), a 12-item self-report instrument, to assess the participants’ levels of perceived sexual stigma outside the military service, including that received from family and friends. The response scale of the HRSS ranges from *strongly disagree* (score 1) to *strongly agree* (score 4). Higher HRSS scores indicate higher levels of perceived stigmatizing attitudes from family members toward participants’ sexual orientation; this present study employed a summed HRSS score [24]. Studies have validated the validity and reliability of the HRSS, including the Taiwanese version [25,26].

#### 2.2.3. Depression

This study used the Center for Epidemiologic Studies Depression Scale (CES-D), a 20-item self-report instrument, to assess the participants’ levels of depression in the preceding month. The CES-D response scale ranges from *rarely or none of the time* (less than 1 day; score 0) to *most or all of the time* (5–7 days; score 4). Higher CES-D scores indicate higher levels of depression; this present study used a summed CES-D score [27]. Studies have demonstrated the validity and reliability of the CES-D, including its Taiwanese version [28,29].

#### 2.2.4. Loneliness

The UCLA Loneliness Scale (UCLA), a 20-item self-report instrument, was employed to assess the participants’ levels of loneliness. The UCLA response scale ranges from *never* (score 1) to *always* (score 4). Higher UCLA scores indicate higher levels of loneliness; this present study used a summed UCLA score [30]. Studies have demonstrated the validity and reliability of the UCLA, including its Taiwanese version [31,32].

#### 2.2.5. Suicidal Ideation

The suicide module of the Mini-International Neuropsychiatric Interview (MINI) was used to assess the participants’ level of suicidal ideation in the preceding month [33]. Each item was answered with a “yes” or “no” response. The total number of items with a “yes” response indicated the severity of suicidal ideation. The validity and reliability of the MINI, including its Taiwanese version, was demonstrated to be satisfactory [34].

### 2.3. Data Analysis

This study calculated descriptive statistics, including the mean (SD) and frequency (percentage), for demographic characteristics, scale scores, and answers on the ESSESiMS. Exploratory factor analysis (EFA) with the principal axis method of factor extraction was then implemented to determine the factor structure of the ESSESiMS. In the EFA, the number of factors was decided by how many factors extracted with an eigenvalue larger than 1. Subsequently, the internal consistency of the ESSESiMS was analyzed using McDonald’s ω, and a ω value > 0.7 indicated adequate internal consistency [35]. Finally, Pearson correlation coefficients were implemented to examine the association of the ESSESiMS with other external measures (i.e., HRSS, CES-D, UCLA, and suicidality) and to demonstrate its concurrent validity. Statistical analyses were primarily conducted using IBM SPSS 20.0; the McDonald’s ω was calculated using the *psych* package in R software [36]. In order to tackle the issue of heterogeneity in the present sample (i.e., different ages, educational levels, sexual orientation, duration of military service, and transgender identification), we examined if the ESSESiMS performs differently in the aforementioned demographic factors. Pearson correlations were used for continuous factors (e.g., age) and mean difference tests (i.e., independent *t*-tests and analysis of variance) were used for categorical factors (e.g., sexual orientation). Moreover, sensitivity analysis and stratification analysis were performed if differences were found in the heterogeneity.

## 3. Results

The participants’ scores for HRSS, CES-D, UCLA, and the MINI scale are presented in Table 2.

Because there were only two participants identifying themselves as transgender, we have removed them from the psychometric testing analyses. However, the psychometric testing on the entire sample (N = 399) is presented in Appendix A (Table A1 and Table A2), and the results were almost identical regardless of including or removing the two participants with a transgender identity. Moreover, age (r = −0.092; *p* = 0.068), duration in military service (r = 0.012; *p* = 0.810), and educational level (F = 1.12; *p* = 0.327) were not significantly associated with the ESSESiMS score. However, participants with a bisexual orientation had a significantly higher ESSESiMS score than those with a gay orientation (M = 0.66 vs. 0.35; t = 2.62; *p* = 0.009). Therefore, the following psychometric testing analyses were conducted for the entire sample without those with a transgender identity first. Then, the analyses were repeated again with a stratified sexual orientation (n = 339 for participants with a gay orientation and 58 for a bisexual orientation).

The EFA results indicated that the ESSESiMS had a single-factor structure (eigenvalue = 2.20 for the first factor; eigenvalue = 0.91 for the second factor). Moreover, the item properties together with the factor loadings of the ESSESiMS items are presented in Table 3. The most frequently reported ESS experience was *ever been verbally harassed or socially excluded in the military* (n = 85 [21.4%]), and the least frequently reported experience was *ever been attacked* (e.g., pushing, kicking, slamming, or other physical attack; n = 6 [1.5%]). In total, 97 (24.4%) participants reported experiencing at least one type of ESS in the military service. The factor loadings ranged between 0.38 and 0.78; with regard to internal consistency, the McDonald’s ω was larger than 0.7 (i.e., 0.75). When stratifying the participants into two subsamples (i.e., gay and bisexual orientation), the single-factor structure remains for both subsamples (eigenvalue = 1.92 for the gay sample and 2.90 for the bisexual sample). However, the gay sample had one item (i.e., *ever been attached*) with a somewhat low factor loading at 0.23 and a slightly low McDonald’s ω at 0.69. For the bisexual sample, all factor loadings were strong (range between 0.47 and 0.86) with good McDonald’s ω at 0.87 (Table 3).

With regard to the concurrent validity of the ESSESiMS (Table 4), its total score was significantly associated with the scores of the HRSS (r = 0.189; *p* < 0.001), CES-D (r = 0.183; *p* < 0.001), UCLA (r = 0.223; *p* < 0.001), and suicidality (r = 0.158; *p* = 0.002). Moreover, the significant associations maintained significance when stratifying the sample into the gay sample (r = 0.114 to 0.213; *p* < 0.001 to 0.036) and the bisexual sample (r = 0.277 to 0.425; *p* < 0.001 to 0.035).

## 4. Discussion

The EFA results indicate that the ESSESiMS should have a single-item structure. The ESSESiMS exhibited an acceptable internal consistency and concurrent validity. Incidents of ESS in the ESSESiMS were significantly associated with perceived sexual stigma outside the military service and with higher levels of depression, loneliness, and suicidal ideation.

Studies on sexual orientation-based harassment in the military have acknowledged that the items on the ESSESiMS accurately assess the nature and level of victimization faced by an individual [4,5,19]. For example, the SOHS also contained the items assessing LGB servicemembers’ experiences of verbal harassment, physical attack, and sexual harassment [19], indicating that these types of ESS are prevalent in the military service across regions. However, no participants of the focus groups in the first stage of this study reported witnessing or experience of graffiti, vandalism, and limited career opportunities contained in the SOHS [19], indicating that the presentations of ESS in the military service have sociocultural and systemic differences. The ESSESiMS also has new items on the respondent’s experiences of supervisor discrimination, including being excluded from receiving help from one’s supervisors and being assigned work unfairly due to their sexual orientation. Delegating work unfairly to sexual minorities constitutes a covert ESS that is not easily verified. Support from supervisors is a crucial element of fair treatment, particularly in closed environments such as that in the military. LGB servicemembers who encounter difficulties but are shunned by supervisors due to their sexual orientation experience feelings of helplessness and hopelessness.

This present study identified verbal harassment and social exclusion as the most common forms of ESS in the military service. The result was similar to that of the RAND study on LGB servicemembers in the U.S. in 2010 [5]. However, verbal harassment and social exclusion may involve more subtle behavior compared with physical attack and sexual harassment [37]. The self-reported experiences on the ESSESiMS can help commanders foster a safe climate for sexual minorities in the military. This present study also found that bisexual participants reported a higher ESS in the military service compared with gay participants. Studies have also reported that bisexual men experienced more social and internalized sexual stigma compared with gay men [38,39]. Compared with the RAND study [5], the participants of this present study reported fewer experiences of physical injuries such as pushing, kicking, slamming, or other physical attack (1.5% vs. 7%). Further study is needed to investigate the reasons accounting for the difference.

This present study determined that experiences of ESS were significantly associated with depression, loneliness, and suicidal ideation in gay and bisexual servicemembers. Experiences of ESS during military service may directly or indirectly affect servicemembers’ cognition, coping skills, emotional regulation, social interaction, and mental health [40]. Although this cross-sectional study did not determine the temporal relationship between ESS and mental health, the prevalence of ESS in the military requires active interventions to ensure the mental health of gay and bisexual servicemembers.

These results highlight the necessity of the prevention and early detection of ESS incidents and related adverse mental health outcomes in LGB servicemembers. Research has suggested that creating a military culture that accepts and integrates LGB servicemembers is the fundamental strategy to reducing ESS in the military [7]. Studies have indicated that sexual orientation-based discrimination is underreported [3,41]. The military should encourage LGB individuals who experience harassment and bystanders to report incidents of ESS, ensure the confidentiality and safety of the informers, and provide timely interventions for ESS deterrence. The self-reported ESSESiMS is a valuable instrument for recording incidents of ESS among LGB servicemembers. Healthcare providers should receive additional training regarding the healthcare of LGB servicemembers, thus better enabling clinicians to assist sexual minorities in managing health problems related to ESS [42]. Research has noted that many LGB servicemembers felt uncomfortable disclosing their sexual orientation to healthcare providers in the military [43].

This study has several limitations. First, because this study recruited participants through an online advertisement, selection bias may occur. Second, this study included only GBM. Whether these results can be generalized to lesbian and bisexual women requires further investigation. Third, most of the participants have been discharged from the military. This study examined the levels of current but not in-service depression, loneliness, and suicidality; therefore, it limited inferences regarding how the relationship between ESS in the military and adverse mental health outcomes change over time. Fourth, all data were self-reported by participants, and the researchers could not fully control for single-rater and recall biases. Finally, gender orientation might be a confounder for the results of this study. Specifically, the distribution between gender orientations was skewed (85.5% in the gay sample and 14.5% in the bisexual sample). Therefore, the unbalanced distribution might contribute to the bias in the psychometric findings of the ESSESiMS. Indeed, our stratified analyses showed that the two subgroups had somewhat different features in the psychometric results of the ESSESiMS. Future studies may want to enlarge the sample size and reevaluate the psychometric properties of the ESSESiMS.

## 5. Conclusions

The results of our study supported the psychometric properties of the newly developed ESSESiMS for assessing the experiences of ESS among gay and bisexual servicemembers in Taiwan. Experiences of ESS toward gay and bisexual servicemembers were not uncommon; ESS was also significantly associated with adverse mental health outcomes. Therefore, urgent interventions are required to create a military culture that accepts and integrates LGB servicemembers and to prevent adverse mental health outcomes related to ESS.

## Figures and Tables

**Table 1 ijerph-20-01260-t001:** Participants’ characteristics.

Variables	First Stage (N = 12)	Second Stage (N = 12)
	Mean (SD) or n (%)	Mean (SD) or n (%)
Age (year)	33.21 (2.24)	32.68 (6.72)
Sexual orientation		
Gay	10 (83.3)	341 (85.5)
Bisexual	2 (16.7)	58 (14.5)
Transgender ^a^	0	2 (0.5)
Educational level		
High school or below	1 (8.3)	41 (10.3)
Undergraduate	10 (83.3)	289 (72.4)
Postgraduate	1 (8.3)	69 (17.3)
Duration in military service (month)	14.82 (10.91)	15.42 (23.70)

^a^ Characteristics of two transgender participants: age: 26 and 33 years; sexual orientation: both are gay; educational level: both are undergraduate; duration in military service: 6 and 10 months.

**Table 2 ijerph-20-01260-t002:** Participants’ perceived sexual stigma outside military service, depression, loneliness, and suicidality (N = 399).

Variables	Mean (SD) or n (%)
Sexual stigma on the HRSS	26.8 (6.77)
Depression on the CES-D	16.95 (10.37)
Loneliness on the UCLA	52.70 (4.98)
Suicidality	0.82 (1.23)

HRSS = Homosexuality related stigma scale; CES-D = Center for Epidemiologic Studies Depression Scale; UCLA = UCLA Loneliness Scale.

**Table 3 ijerph-20-01260-t003:** Exploratory factor analysis results of the Enacted Sexual Stigma Experiences Scale in Military Service (ESSESiMS).

Item Description: Because of Your Gay or Bisexual Identify, Have the Following Descriptions Ever Happened to You in Your Military Service?	Response of Yes (%)	Factor Loading
Sample without those with transgender identification (n = 397) ^a^		
1. Ever been verbally harassed or socially excluded	85 (21.4)	0.53
2. Ever been attacked (e.g., pushed, kicked, slammed, or other physical attack)	6 (1.5)	0.38
3. Ever been tortured by unfair work assignment	22 (5.5)	0.78
4. Ever been sexually harassed or forced into sexual behaviors	23 (5.8)	0.45
5. Ever been rejected from receiving supervisors’ help	20 (5.0)	0.59
Gay sample without those with transgender identification (n = 339) ^b^		
1. Ever been verbally harassed or socially excluded	70 (20.6)	0.61
2. Ever been attacked (e.g., pushed, kicked, slammed, or other physical attack)	4 (1.2)	0.23
3. Ever been tortured by unfair work assignment	15 (4.4)	0.68
4. Ever been sexually harassed or forced into sexual behaviors	15 (4.4)	0.37
5. Ever been rejected from receiving supervisors’ help	15 (4.4)	0.47
Bisexual sample without those with transgender identification (n = 58) ^c^		
1. Ever been verbally harassed or socially excluded	15 (25.9)	0.47
2. Ever been attacked (e.g., pushed, kicked, slammed, or other physical attack)	2 (3.4)	0.66
3. Ever been tortured by unfair work assignment	7 (12.1)	0.86
4. Ever been sexually harassed or forced into sexual behaviors	8 (13.8)	0.60
5. Ever been rejected from receiving supervisors’ help	6 (10.3)	0.84

^a^ Eigenvalue = 2.20; Explained variance = 44.00%; McDonald’s ω = 0.75. ^b^ Eigenvalue = 1.92; Explained variance = 38.46%; McDonald’s ω = 0.69. ^c^ Eigenvalue = 2.90; Explained variance = 57.92%; McDonald’s ω = 0.87.

**Table 4 ijerph-20-01260-t004:** Concurrent validity of the Enacted Sexual Stigma Experiences Scale in Military Service (ESSESiMS).

			R (*p*-Value)		
	ESSESiMS	HRSS	CES-D	UCLA	Suicidality
Sample without those with transgender identification (n = 397)					
ESSESiMS	--				
HRSS	0.187 (<0.001)	--			
CES-D	0.181 (<0.001)	0.310 (<0.001)	--		
UCLA	0.224 (<0.001)	0.212 (<0.001)	0.425 (<0.001)	--	
Suicidality	0.157 (0.002)	0.136 (0.007)	0.459 (<0.001)	0.099 (0.048)	--
Gay sample without those with transgender identification (n = 339)					
ESSESiMS	--				
HRSS	0.174 (0.001)	--			
CES-D	0.124 (0.023)	0.284 (<0.001)	--		
UCLA	0.213 (<0.001)	0.198 (<0.001)	0.407 (<0.001)	--	
Suicidality	0.114 (0.036)	0.116 (0.032)	0.455 (<0.001)	0.084 (0.124)	--
Bisexual sample without those with transgender identification (n = 58)					
ESSESiMS	--				
HRSS	0.318 (0.015)	--			
CES-D	0.425 (0.001)	0.450 (<0.001)	--		
UCLA	0.277 (0.035)	0.316 (0.016)	0.529 (<0.001)	--	
Suicidality	0.386 (0.003)	0.268 (0.042)	0.490 (<0.001)	0.208 (0.118)	--

HRSS = Homosexuality related stigma scale; CES-D = Center for Epidemiologic Studies Depression Scale; UCLA = UCLA Loneliness Scale.

## Data Availability

The data used in this study are available upon reasonable request to the corresponding authors.

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
