# Peer review of "Assessing Enacted Sexual Stigma toward Gay and Bisexual Men in the Military: The Enacted Sexual Stigma Experiences Scale in Military Service"

_ijerph, 2023, doi:10.3390/ijerph20021260_

Round 1

Reviewer 1 Report

A very interesting paper, about a very actual topic and well written. I would to make some comments to the authors, in order to increase the chances for publication of the paper:

- Lines 44-53 are a little bit confusing for me, especially with the style of redaction. Please, I would like to recomment the revission of these lines, mainly the way to introduce previous studies about the topic.

- Lines 90-92 are similar to lines 54-55, I suggest to make a change in the redaction.

- The aims of the study are not clearly presented, neither at the end of Introduction nor and the beginning of Methods. I suggest to include the aims in one of these parts of the text.

- In lines, 100-103, the authors express: "The study excluded
individuals with any conditions that prevented them from completing the questionnaire, such as those with cognitive impairment or dysfunction due to major physical or psychiatric disorders". Can you briefly explain how you made this split? It's  not clear in the text.

- In lines 126-128, the authors express: "The present study asked the participants to report their age, educational level, sexual
orientation, and duration of military service and whether they identified as being transgender". There are no data about these factors and at less two of them can be really significant for the study: age (and if there are differencies in experienced ESS ky age, wich it can indicate an evolution in the situation) and trans identity (wich can absolutely change the categories of the analysis). Please, explain how these factors have been analyzed and the influence in the validation of the questionnaire.

- About the Results, there is a enormous difference between the groups of analysis (85%/15%), so it's really difficult to know if the validation of the questionnaire is correct for both groups. Please, explain how the authors have worked this aspect.

- Related with the previous comment, the Results are presented with the sample as an homogeneus group, what it's not the proper way of analysis. At less, it's necessary to make a global analysis and a split between G-B groups to determinate the validity and feasibility of the questionnaire for both groups.

- As it has been expressed previously, the difference between collectives (G-B) is a bias and it has to be included in the limitations part.

Author Response

We appreciated your valuable comments. As discussed below, we have revised our manuscript with underlines based on your suggestions. Please let us know if we need to provide anything else regarding this revision.

Comment 1

- Lines 44-53 are a little bit confusing for me, especially with the style of redaction. Please, I would like to recommend the revision of these lines, mainly the way to introduce previous studies about the topic.

Response

Thank you for your comment. We revised the contents regarding the introduction of previous studies about the topic as below. Please refer to line 44-56.

Enacted sexual stigma (ESS) is a type of sexual stigma characterized by overt discriminatory behaviors against lesbian, gay, and bisexual (LGB) individuals [1]. LGB military servicemembers experience disproportionately high rates of victimization due to ESS [2,3]. For example, a study on 71,570 servicemembers in the active-duty military of the United States (U.S.) in 2000 revealed that 37% had witnessed or experienced harassment and violence based on perceptions or suspicions that the victim was gay; physical assault was reported by 5.3% of respondents [4]. Another U.S. study on 208 LGB servicemembers in 2010 reported that 29% of participants have been teased or mocked and 7% have received threats or injuries because of their sexual orientation [5]. In addition, a U.S. study on 253 LGB servicemembers in 2013 also reported that 83% of lesbian and bisexual women and 74% gay and bisexual men experienced sexual harassment or assault in the past 12 months [3]. The results of previous studies have supported that ESS is a very common issue in military service that needs attention.

Comment 2

- Lines 90-92 are similar to lines 54-55, I suggest to make a change in the redaction.

Response

We revised the section introducing the hypotheses of this study as below. Please refer to line 106-112.

According to ecological systems theory [23], the army is an ecological system interacting with other systems. Therefore, it is hypothesized that the ESS measured by the ESSESiMS is significantly associated with perceived sexual stigma outside military services. Moreover, according to minority stress theory [6], ESS constitutes a psychological stressor for GBM and negatively affects their mental health. Therefore, it is hypothesized that the ESS measured by the ESSESiMS is significantly associated with depression, loneliness, and suicidality.

Comment 3

- The aims of the study are not clearly presented, neither at the end of Introduction nor and the beginning of Methods. I suggest to include the aims in one of these parts of the text.

Response

We revised the section introducing the hypotheses of this study as below. Please refer to line 100-106.

The Enacted Sexual Stigma Experiences Scale in Military Service (ESSESiMS) is a newly formulated scale assessing the experience of ESS among gay and bisexual servicemembers in Taiwan. The present study aimed to examine the psychometrics of the ESSESiMS. In addition to examining factor structure of the ESSESiMS, this study also examined the concurrent validity of the ESSESiMS by testing its correlations with perceived sexual stigma outside military services, depression, loneliness, and suicidal ideation in gay and bisexual servicemembers.

Comment 4

- In lines, 100-103, the authors express: "The study excluded individuals with any conditions that prevented them from completing the questionnaire, such as those with cognitive impairment or dysfunction due to major physical or psychiatric disorders". Can you briefly explain how you made this split? It's not clear in the text.

Response

Thank you for your comment. We added the contents explaining how to exclude the individuals with any conditions that prevented them from completing the questionnaire as below. Please refer to line 135-145.

The study excluded individuals with any conditions that prevented them from completing the questionnaire, such as those with cognitive impairment or dysfunction due to major physical or psychiatric disorders. Individuals who were interested in this study could contact research assistants by telephone. Research assistants tentatively evaluated the eligibility of potential participants, explained the study aims and procedures, and scheduled a time for eligible participants to complete the study questionnaires. A total of 401 potential participants were invited to participate in this study. The research assistants further determine whether they had conditions that might compromise their ability to complete the questionnaire in interview rooms of a university-affiliated hospital. Two individuals were excluded due to low mentality or alcohol intoxication.

Comment 5

- In lines 126-128, the authors express: "The present study asked the participants to report their age, educational level, sexual orientation, and duration of military service and whether they identified as being transgender". There are no data about these factors and at less two of them can be really significant for the study: age (and if there are differences in experienced ESS ky age, wich it can indicate an evolution in the situation) and trans identity (wich can absolutely change the categories of the analysis). Please, explain how these factors have been analyzed and the influence in the validation of the questionnaire.

Response

Thank you for the reminder. We have now analyzed if there are significant differences in the scores of Enacted Sexual Stigma Experiences Scale in Military Service (ESSESiMS) between these variables, except for the transgender. We did not analyze for transgender because there were only two participants with transgender. Therefore, we have removed the two participants from the analyses. Nevertheless, we found that the results in psychometric testing remain the same after removing the two participants. Regarding the other demographic factors, we found that age, duration of military service, and educational level were not associated with the ESSESiMS score. However, gay sample and bisexual sample had significantly different ESSESiMS scores. Therefore, we have conducted stratified analyses to examine the psychometric properties of the ESSESiMS for the two subsamples separately. The psychometric results were similar, although slightly different between the two subsamples. Nevertheless, the one-factor structure of the ESSESiMS was supported for both subsamples. We have now added the information in the revised manuscript (also please see the revised Tables 3 and 4 in the revised manuscript).

“In order to tackle the issue of heterogeneity in the present sample (i.e., different ages, educational levels, sexual orientation, duration of military service, and transgender identification), we examined if ESSESiMS performs differently in the aforementioned demographic factors. Pearson correlations were used for continuous factors (e.g., age) and mean difference tests (i.e., independent t-tests and analysis of variance) were used for categorical factors (e.g., sexual orientation). Moreover, sensitivity analysis and stratification analysis were done if differences were found in the heterogeneity.” Please refer to line 201-208.

“Because there were only two participants identifying themselves as transgender, we have removed them from the psychometric testing analyses. However, the psychometric testing on the entire sample (N = 399) is presented in the Appendix, and the results were almost identical regardless including or removing the two participants with transgender identity. Moreover, age (r = -0.092; p = .068), duration in military service (r = 0.012; p = .810), educational level (F = 1.12; p = .327) were not significantly associated with ESSESiMS score. However, participants with bisexual orientation had significantly higher ESSESiMS score than those with gay orientation (M = 0.66 vs. 0.35; t = 2.62; p = .009). Therefore, the following psychometric testing analyses were conducted for the entire sample without those with transgender identity first. Then, the analyses were repeated again with stratified sexual orientation (n = 339 for participants with gay orientation and 58 for bisexual orientation).” Please refer to line 216-227.

“When stratifying the participants into two subsamples (i.e., gay and bisexual orientation), the single-factor structure remains for both subsamples (eigenvalue = 1.92 for gay sample and 2.90 for bisexual sample). However, the gay sample had one item (i.e., ever been attached) with a somewhat low factor loading at 0.23 and the a slightly low McDonald’s ω at 0.69. For the bisexual sample, all factor loadings were strong (range between 0.47 and 0.86) with good McDonald’s ω at 0.87 (Table 3).” Please refer to line 236-242.

“Moreover, the significant associations maintained significant when stratifying the sample into gay sample (r = 0.114 to 0.213; p <.001 to .036) and bisexual sample (r = 0.277 to 0.425; p <.001 to .035).” Please refer to line 250-253.

Comment 6

- About the Results, there is an enormous difference between the groups of analysis (85%/15%), so it's really difficult to know if the validation of the questionnaire is correct for both groups. Please, explain how the authors have worked this aspect. Related with the previous comment, the Results are presented with the sample as an homogeneus group, what it's not the proper way of analysis. At less, it's necessary to make a global analysis and a split between G-B groups to determinate the validity and feasibility of the questionnaire for both groups.

Response

Thank you. We have now analyzed additional psychometric testing for the two subgroups. Please see detailed information in our response to your previous comment.

Comment 7

- As it has been expressed previously, the difference between collectives (G-B) is a bias and it has to be included in the limitations part.

Response

We have now added this as one of the limitations.

Finally, the gender orientation may be an important confounder for the present study results. Specifically, the distribution between the gender orientations was skewed (85.5% in gay sample and 14.5% in bisexual sample). Therefore, the unbalanced distribution may contribute to bias in the psychometric findings of the ESSESiMS. Indeed, our stratified analyses showed that the two subgroups had somewhat different features in the psychometric results of the ESSESiMS. Future studies may want to enlarge the sample size and reevaluate the psychometric properties of the ESSESiMS.” Please refer to line 321-327.

Reviewer 2 Report

The authors address a topic, that is fairly covered in the literature -if one browses Sexual Stigma Toward Gay and Bisexual Men in the Military” Google gives 676,000 results in 0.52 seconds - however the short paper focuses on a country, Taiwan which is underrepresented in the literature and is reasonably well structured.

There are some minor points that the authors should, however, enhance in order to better present the arguments and the findings, as indicated in the following points.

Introduction: Although the authors have indicated that there are few studies on Sexual Stigma Toward Gay and Bisexual Men in the Military outside the USA and European contexts, they should still try to extend the literature review, and to provide some more examples on the impact of ESS among the studied group; this would offer the reader with a clearer understanding of the relevance of the phenomenon investigated.

It should also be explained the correlation/causality between the sexual stigma experience in military service and that experienced outside the military services otherwise the rationale of section 2.2.3 it is not justified. On the same line, it should be indicated if depression, loneliness and suicidal attitude were reported during the military service and/or outside of it

Materials and Methods: The authors should expand this part explaining the type of methodology used followed by the research instruments. The different steps followed to obtain the relevant information should also properly described. In particular it should be explained the different roles played by the 399 GBM who were interview via questionnaire and the 12 GBM who were interview via focus groups and the reason of having two focus groups with, apparently the same group of individuals, as well as the reason of using different form of recruitments for the participants at the focus groups.

The division of roles among the investigators and the research assistants should be better explained, avoiding expression such as “we” (line 112).

Data analysis: For the sake of clarity table 1 could be divided into two parts, a first table with the participants' demographic characteristics (Age, Sexual orientation, Educational level, and Duration in military service) which could be moved to section 2.2.2., and a second table with the remaining explanatory elements. At the same time, it would be interesting if in the discussion the authors would divulge on describing if for example, age, educational level, and duration in military service, do have an impact on the type of stigma experiences.

This section would also benefit from cross-referencing from similar studies, so to highlight the differences and/or similarities  

Author Response

We appreciated your valuable comments. As discussed below, we have revised our manuscript with underlines based on your suggestions. Please let us know if we need to provide anything else regarding this revision.

Comment 1

Introduction: Although the authors have indicated that there are few studies on Sexual Stigma Toward Gay and Bisexual Men in the Military outside the USA and European contexts, they should still try to extend the literature review, and to provide some more examples on the impact of ESS among the studied group; this would offer the reader with a clearer understanding of the relevance of the phenomenon investigated.

Response

Thank you for your comment. We added a new paragraph to introduce the relationship between ESS and mental health problems in military servicemembers as below. Please refer to line 64-71.

Research has found that sexual orientation-based harassment is significantly associated with decreased social cohesion among U.S. military veterans [16]; decreased social support can further increase the risk of suicidal ideation [17]. Sexual assault during military service is significantly associated with the risk of post-traumatic stress disorder (PTSD) in lesbian and bisexual female veterans [18]. Conceal of sexual orientation is a common method for LGB servicemembers to avoid ESS in military service; however, anxiety around concealment of one’s sexual orientation while in the service was related to current depression and PTSD symptoms [14].

Comment 2 

It should also be explained the correlation/causality between the sexual stigma experience in military service and that experienced outside the military services otherwise the rationale of section 2.2.3 it is not justified. On the same line, it should be indicated if depression, loneliness and suicidal attitude were reported during the military service and/or outside of it

Response

Thank you for your comment. In the revised manuscript, we added the contents introducing the hypotheses regarding examining the associations of ESS with perceived sexual stigma outside military service, depression, loneliness, and suicidality. Furthermore, this study collected participants’ current perceived sexual stigma outside military service, depression, loneliness, and suicidality. However, most of participants have discharged from the military, and thus this study could not determine the temporal relationships of ESS with depression, loneliness, and suicidality. We listed it as one of the limitations of this study.

According to ecological systems theory [23], the army is an ecological system interacting with other systems. Therefore, it is hypothesized that the ESS measured by the ESSESiMS is significantly associated with perceived sexual stigma outside military services. Moreover, according to minority stress theory [6], ESS constitutes a psychological stressor for GBM and negatively affects their mental health. Therefore, it is hypothesized that the ESS measured by the ESSESiMS is significantly associated with depression, loneliness, and suicidality. Please refer to line 106-112.

Third, most of participants have discharged from the military. This study examined the levels of current but not in-service depression, loneliness, and suicidality; therefore, it limited inferences regarding how the relationship between ESS in the military and adverse mental health outcomes change over time. Please refer to line 316-319.

Comment 3

Materials and Methods: The authors should expand this part explaining the type of methodology used followed by the research instruments. The different steps followed to obtain the relevant information should also properly described. In particular it should be explained the different roles played by the 399 GBM who were interview via questionnaire and the 12 GBM who were interview via focus groups and the reason of having two focus groups with, apparently the same group of individuals, as well as the reason of using different form of recruitments for the participants at the focus groups.

Response

Thank you for your comment. In the revised manuscript, we revised the contents of “2.1. Participants and Procedure” and integrate the participants and procedure of study stage 1 for developing the scale and stage 2 for validating the scale as below. Please refer to line 115-146. We also examined the differences in age, sexual orientation, education level and duration in military service between the participants of two stages. We found that no significant differences existed. Please refer to line 152-157.

“The present study took a two-stage approach to develop the ESSESiMS and examine its psychometric characteristics. Firstly, the investigators collected data on incidents of ESS in miliary service from 12 GBM in two focus groups interviews to help develop the ESSESiMS in January and February 2021. This study recruited participants by posting advertisements on the home pages of three online LGB communities. GBM with at least 4 months of military service experience in Taiwan were eligible to participate in the focus group interviews. The investigators determined the discussion topics and led the group discussion on incidents of ESS that the participants had experienced or witnessed during military service. Two researchers reviewed the transcript and coded the data for ESS in military service. The principal investigator reviewed the coding results and integrated them into the following five items of the ESSESiMS: verbal harassment or social exclusion, physical violence (e.g., pushing, kicking, slamming, or other physical attack), unfair assignment of work, sexual harassment or abuse, and being rejected from receiving supervisors’ help. Each item was answered with “yes” or “no.”

Secondly, this study recruited participants to validate the ESSESiMS through advertisements on social media (Facebook, LINE, and Twitter) and on a popular forum in Taiwan, called the Bulletin Board System, from August 1, 2021, to May 1, 2022. The study included Taiwanese men who 1) identified as gay or bisexual, 2) were ≥20 years of age, and 3) were currently completing or had previously completed miliary service in Taiwan. The study excluded individuals with any conditions that prevented them from completing the questionnaire, such as those with cognitive impairment or dysfunction due to major physical or psychiatric disorders. Individuals who were interested in this study could contact research assistants by telephone. Research assistants tentatively evaluated the eligibility of potential participants, explained the study aims and procedures, and scheduled a time for eligible participants to complete the study questionnaires. A total of 401 potential participants were invited to participate in this study. The research assistants further determine whether they had conditions that might compromise their ability to complete the questionnaire in interview rooms of a university-affiliated hospital. Two individuals were excluded due to low mentality or alcohol intoxication. A total of 399 eligible GBM granted informed consent and individually completed questionnaires.”

There were no significant differences in age, sexual orientation, education level and duration in military service between the participants of the first and second stages. No participant in the first stage had transgender identity.” and Table 1.

Comment 4

The division of roles among the investigators and the research assistants should be better explained, avoiding expression such as “we” (line 112).

Response

We replaced “we” by “the investigators.” Please refer to line 116.

Comment 5

Data analysis: For the sake of clarity table 1 could be divided into two parts, a first table with the participants' demographic characteristics (Age, Sexual orientation, Educational level, and Duration in military service) which could be moved to section 2.2.2., and a second table with the remaining explanatory elements. At the same time, it would be interesting if in the discussion the authors would divulge on describing if for example, age, educational level, and duration in military service, do have an impact on the type of stigma experiences.

Response

Thank you for your comment. Accordingly, we divided Table 1 into “Table 1. Participants’ characteristics” (line 155) and “Table 2. Participants’ perceived sexual stigma outside military service, depression, loneliness, and suicidality (N = 399)” (line 212).

Comment 6

This section would also benefit from cross-referencing from similar studies, so to highlight the differences and/or similarities  

Response

We added new contents into Discussion section to compare the results of this study with those of previous studies as below. Please refer to line 263-290.

Studies on sexual orientation–based harassment in the military have acknowledged that the items on the ESSESiMS accurately assess the nature and level of victimization faced by an individual [4,5,19]. For example, the SOHS also contained the items assessing LGB servicemembers’ experiences of verbal harassment, physical attack, and sexual harassment [19], indicating that these types of ESS are prevalent in military service across regions. However, no participants of focus groups in the first stage of study reported witlessness or experience of graffiti, vandalism, and limited career opportunities contained in the SOHS [19], indicating that presentations of ESS in military service have sociocultural and systemic differences. The ESSESiMS also has new items on the respondent’s experiences of supervisor discrimination, including being excluded from receiving help from one’s supervisors and being assigned work unfairly due to their sexual orientation. Delegating work unfairly to sexual minorities constitutes a covert ESS that is not easily verified. Support from supervisors is a crucial element of fair treatment, particularly in closed environments such as that in the military. LGB servicemembers who encounter difficulties but are shunned by supervisors due to their sexual orientation experience feelings of helplessness and hopelessness.

The present study identified verbal harassment and social exclusion as the most common forms of ESS in military service. The result was similar to that of the RAND study on the U.S. LGB servicemembers in 2010 [5]. However, verbal harassment and social exclusion may involve more subtle behavior compared with physical attack and sexual harassment [37]. The self-reported experiences on the ESSESiMS can help commanders foster a safe climate for sexual minorities in the military. The present study also found that bisexual participants reported higher ESS in military service compared with gay participants. Studies have also reported that bisexual men experienced more social and internalized sexual stigma compared with gay men [38,39]. Compared with the RAND study [5], the participants of the present study reported fewer experiences of physical injuries such as such as push, kick, slam, or other physical attack (1.5% vs. 7%). Further study is needed to investigate the reasons accounting for the difference.

Round 2

Reviewer 1 Report

Lot of thanks for your efforts to include the comments and suggestions in the paper, I really appreciate the work of the authors. All the improvements are in the text now, good luck with your future research about this very actual and interesting topic.